# Intranasal Administration of Acetaminophen-Loaded Poly(lactic-*co*-glycolic acid) Nanoparticles Increases Pain Threshold in Mice Rapidly Entering High Altitudes

**DOI:** 10.3390/pharmaceutics17030341

**Published:** 2025-03-06

**Authors:** Qingqing Huang, Xingyue Han, Jin Li, Xilin Li, Xin Chen, Jianwen Hou, Sixun Yu, Shaobing Zhou, Gu Gong, Haifeng Shu

**Affiliations:** 1Department of Neurosurgery, The General Hospital of Western Theater Command, College of Medicine, Southwest Jiaotong University, Chengdu 610031, China; corn1993@my.swjtu.edu.cn (Q.H.); jianxin@my.swjtu.edu.cn (X.H.); chenxin0852@hotmail.com (X.C.); ysx1982@aliyun.com (S.Y.); 2College of Medicine, Southwest Jiaotong University, Chengdu 610031, China; lijin@uok.edu.gr (J.L.); xilinli@swjtu.edu.cn (X.L.); houjianwen@swjtu.edu.cn (J.H.); shaobingzhou@swjtu.edu.cn (S.Z.); 3Department of Anesthesiology, The General Hospital of Western Theater Command, Chengdu 610031, China; gonggu68@uok.edu.gr

**Keywords:** acetaminophen, poly(lactic-*co*-glycolic acid), nanoparticle, intranasal administration, plateau, pain threshold, high altitude, drug encapsulation, nanocarrier, mouse model

## Abstract

**Background/Objectives:** Orally or intravenously administered acetaminophen experiences considerable liver first-pass elimination and may cause liver/kidney damage. This work examined the pharmacological effects of acetaminophen-loaded poly(lactic-*co*-glycolic acid) nanoparticles (AAP PLGA NPs) intranasally administered to mice rapidly entering high altitudes. **Methods:** AAP PLGA NPs were prepared using ultrasonication-assisted emulsification and solvent evaporation and characterized in terms of drug encapsulation efficiency and loading, in vitro and in vivo release behaviors, and toxicity to hippocampal neurons. In vivo fluorescence imaging was used to monitor the concentrations of AAP PLGA NPs (labeled with indocyanine green) in the brain and blood of the mice after intranasal administration. The effects of these NPs on the pain threshold in mice rapidly entering high altitudes were evaluated through hot plate and tail flick experiments. **Results:** The AAP PLGA NPs were found to be noncytotoxic, highly biocompatible and stable, with a drug encapsulation efficiency and loading capacity of 42.53% and 3.87%, respectively. The in vitro release of acetaminophen lasted for up to 72 h, and the release rate was ~82%. After intranasal administration in vivo, the drug release occurred slowly, and the drug was mainly concentrated in the brain. Compared with nonencapsulated acetaminophen, the intranasal administration of AAP PLGA NPs resulted in higher brain levels of the drug and delayed its elimination, thus increasing the pain threshold in mice rapidly entering high altitudes. **Conclusions:** The proposed strategy addresses the common problems of intranasal drug administration (low retention time and bioavailability) and paves the way for effective pain management in high-altitude environments.

## 1. Introduction

As tourism in high-altitude regions becomes increasingly popular, the impact of high-altitude environments on human health is being extensively studied. Such environments typically have altitudes of >3000 m and are characterized by low oxygen levels, low air pressures, low temperatures, dryness, and strong ultraviolet radiation, which strongly affect the human digestive, respiratory, circulatory, and nervous systems [1,2,3]. Pain experienced at high altitudes can affect various aspects of human physiology, psychology, and life quality, which can result in increased pain sensitivity, prolonged pain duration, expanded pain range, anxiety, depression, emotional instability, restricted daily activities, and poor sleep quality [4].

Acetaminophen is a commonly used antipyretic and analgesic that inhibits prostaglandin synthesis in the central nervous system and is suitable for the treatment of headaches and muscle aches caused by altitudinal sickness [5,6]. This drug is mainly administered orally and intravenously and may cause liver and kidney damage upon high dosage or long-term use [7]. Intranasal administration is a viable alternative to oral and intravenous administration, offering the advantages of noninvasiveness, ease of operation, rapid absorption and action onset, and avoidance of liver first-pass elimination [8]. However, the physiological environment of the nasal cavity, including the mucociliary clearance system and enzyme barrier of the nasal mucosa, can limit the retention time and bioavailability of intranasally administered drugs [9]. When researching and applying intranasal drug delivery, one should consider these limitations and take appropriate measures to overcome/reduce their impact, for example, by developing drug carrier systems such as liposomes, nanoemulsions, and nanoparticles (NPS) [9].

With the development of nanotechnology and optimization of drug delivery systems, the application scope of PLGA-based nanodrug delivery systems has expanded to include intranasal administration [10]. Owing to its good biocompatibility and degradability, PLGA can be degraded in vivo and excreted from the body, which helps avoid issues related to the long-term persistence of drug carrier residues [11]. By adjusting the composition and size of PLGA NPs, one can achieve controlled drug release and thus prolong the retention time of drugs in the nasal cavity, improve drug bioavailability, reduce the frequency of drug administration, and protect the drugs from hydrolysis by enzymes in the nasal cavity to enhance drug stability [12].

Therefore, in this study, we aimed to develop an efficient method of pain management in high-altitude environments. We explored the effects of intranasally administered acetaminophen-loaded PLGA nanoparticles (AAP PLGA NPs) on the pain threshold in mice rapidly entering high altitudes.

## 2. Materials and Methods

### 2.1. Instruments

The following instruments were used in this study: NanoBrook Omni particle size potential analyzer (Brookhaven, NY, USA); pure water machine (Sichuan Youpu Ultra Pure Technology Co., Ltd., Chengdu, China); DHG-9070A electric constant-temperature blast-drying oven (Shanghai Jinghong Experimental Equipment Co., Ltd., Shanghai, China); S10-3 type constant-temperature magnetic stirrer (Shanghai Sile Instrument Co., Ltd., Shanghai, China); ZWY-103D constant-temperature oscillator (Shanghai Zhicheng Analytical Instrument Manufacturing Co., Ltd., Shanghai, China); LA310s precision electronic balance (Mettler Toledo, Greifensee, Switzerland); UV–visible spectrophotometer (Lambda 25, PerkinElmer, Waltham, MA, USA); JEM-1200EX transmission electron microscope (Jeol Corporation, Tokyo, Japan); S-450D ultrasonic cell disruptor (Branson, Danbury, CT, USA); DMIL-LED fluorescence inverted microscope (Leica, Wetzlar, Germany); TDL-60B desktop centrifuge (Shanghai Anting Scientific Instrument Factory, Shanghai, China); Newton 7.0 FT-500 3D small animal live imaging system (Vilber, France); Multiskan FC ELISA reader (Thermo Fisher Instruments, Waltham, MA, USA); cell culture incubator (Thermo Fisher Instruments, Waltham, MA, USA); low-pressure oxygen chamber (Shanghai Tawang Intelligent Technology Co., Ltd., Shanghai, China); rat tail light pain tester (Jiangsu Saion Biotechnology Co., Ltd., Nanjing, China); hot plate pain tester (Shanghai Xinxin Information Technology Co., Ltd., Shanghai, China).

### 2.2. Drugs and Reagents

The following drugs, reagents, and materials were used in this study: acetaminophen API (Shanghai Aladdin Biochemical Technology Co., Ltd., Shanghai, China, ≥99.5% purity); polylactic acid–hydroxyacetic acid copolymer (Shanghai Yuanye Biotechnology Co., Ltd., Shanghai, China); indocyanine green (ICG, MedChemExpress, Princeton, NJ, USA, ≥98.0% purity); artificial cerebrospinal fluid and artificial nasal fluid (Dongguan Chuangfeng Automation Technology Co., Ltd., Dongguan, China); dialysis bag (Sigma-Aldrich, St. Louis, MO, USA, 3500Da); DMEM medium, trypsin, fetal bovine serum, and PBS buffer (Suzhou Thermo Fisher Instrument Co., Ltd., Suzhou, China); cell culture consumables (Costar, Arlington, VA, USA); CCK-8 detection kit and Calcein AM/PI kit (Beijing Lanjieke Technology Co., Ltd., Beijing, China).

### 2.3. Animals and Cells

Mouse hippocampal neuron HT22 cells were obtained from Wuhan Ponosi Life Technology Co., Ltd. (Wuhan, China), and C57/BL6 mice from Gempharmatech Co., Ltd. (Nanjin, China; license: SCXK (Chuan) 2020-034) were used in the experiments. All experiments strictly followed the 3R principles for the use of experimental animals and provision of humane care.

### 2.4. Preparation of AAP PLGA NPs

Acetaminophen (2 mg) was dissolved in a solution of PLGA (20 mg; 13 kDa, lactic acid:glycolic acid = 75:25, mol/mol) in dichloromethane (1 mL). A beaker containing the resulting oil-phase solution was placed in an ice water bath and ultrasonicated using an ultrasonic cell disruptor (60% power, emulsification time = 45 s, pulse duration = 5 s, interval = 5 s). The resulting emulsion was supplemented with 1 wt% aqueous polyvinyl alcohol (3 mL) and ultrasonicated (60% power, emulsification time = 30 s, pulse duration = 5 s, interval = 5 s) to fragment and stabilize the emulsion particles. Dichloromethane was evaporated through low-speed magnetic stirring at room temperature. Subsequently, the NPs were collected by centrifugation (12,000 rpm, 15 min, 4 °C), washed twice with pure water, and redispersed in a small amount of the same solution. The resulting dispersion was lyophilized in a low-temperature freeze-dryer for 24 h to obtain AAP PLGA NPs (Figure 1A).

### 2.5. Characterization of AAP PLGA NPs

#### 2.5.1. Determination of Drug Loading and Encapsulation Efficiency

The AAP PLGA NPs were centrifuged (12,000 rpm, 20 min, 4 °C) and then washed twice with pure water, and the supernatant was collected in small bottles. The drug loading capacity (DL) and encapsulation efficiency (EE) were determined by UV–visible spectroscopy, and the absorbance was measured at 257 nm.EE = 100% × *m*_packages_/*m*_drug_,(1)DL = 100% × *m*_packages_/*m*_total_,(2)
where *m*_packages_ is the mass of the encapsulated drug (mg), *m*_drug_ is the mass of the administered drug (mg), and *m*_total_ is the mass of the drug-loaded NPs (mg).

#### 2.5.2. Determination of Particle Size and Zeta Potential

The nanoparticle solution was diluted with pure water to an appropriate concentration, and the particle size (diameter), polydispersity index (PDI), and zeta potential were measured using a nanoparticle size (diameter) and zeta potential meter with a fixed scattering angle of 90°. After equilibrating at 25 °C for 2 min, each sample was measured 9 times. The collection time was 1 s, and the average duration of each measurement exceeded 2 min.

#### 2.5.3. Morphological Characterization

The nanoparticle solution was diluted with pure water to an appropriate concentration. One or two drops of the diluted solution were placed on a copper mesh, stained with phosphotungstic acid (10.0 g/L), and air-dried at room temperature. The dried sample was imaged by transmission electron microscopy.

### 2.6. Evaluation of In Vitro Release Behavior and Stability

Given that the AAP PLGA NPs were prepared for intranasal administration and targeted the brain, their in vitro release behavior was characterized in three media (simulated bodily fluids), namely, artificial cerebrospinal fluid (CSF), simulated nasal fluid (SNF), and phosphate-buffered saline (PBS). Acetaminophen raw material and AAP PLGA NPs were accurately weighed into a dialysis bag, and the release medium (5 mL) was added. The other end of the dialysis bag was tightened and immersed in the release medium (20 mL) [13]. The bag was placed in a constant-temperature shaker to initiate dynamic dialysis, which was performed at a temperature of 37 ± 0.5 °C and an oscillation frequency of 100 rpm for 72 h. At a predetermined time (0, 2, 4, 6, 8, 12, 24, 48, or 72 h), a 1 mL aliquot was withdrawn and replaced with an equal volume of fresh release medium (1 mL). The sample was filtered through a 0.45 μm microporous membrane, and the absorbance of the filtrate at 257 nm was used to determine the drug concentration, calculate the cumulative drug release percentage, and construct an in vitro release curve. The release rate was calculated as follows:release percentage = 100% × (drug concentration × medium volume/concentration of active ingredient in the nanomedicine).(3)

An equal amount of AAP PLGA NPs was accurately weighed and placed in three 50 mL centrifuge tubes, to each of which was added 20 mL PBS, CSF, or SNF. At 6, 12, 24, 48 and 72 h, 1 mL of the nanosphere suspension was taken and its particle size, PDI, and zeta potential were measured using the method described under Section 2.5.2.

### 2.7. Cytotoxicity Evaluation

#### 2.7.1. CCK-8 Assay

The HT22 cells were inoculated into a 96-well plate at a density of 3 × 10^4^ cells per well, supplemented with the complete culture medium (89% H-DMEM medium, 10% FBS, 1% AB/AM), and incubated at 37 °C in an atmosphere of 5% CO_2_ for 24 h. Placebo NPs (blank group), AAP sol (control group), or AAP PLGA NPs (treatment group) were added at different concentrations (0.1, 1, 10, and 100 mM), and the cells were incubated for 24 h. The original culture medium was removed, and the cells were washed twice with PBS. Each well was supplemented with a mixture of the CCK-8 reagent (10 μL) and basic culture medium (90 μL). After incubation in the dark for 2 h, an enzyme-linked immunosorbent assay reader was used to measure the optical density (OD) of each well at 450 nm. The cell survival rate was calculated as follows:Cell survival rate = 100% × (OD_treatment group_ − OD_blank group_)/(OD_control group_ − OD_blank group_).(4)

#### 2.7.2. Live/Dead Cell Staining

Live/dead cell assay kits were used to examine biocompatibility. The HT22 cells were inoculated in a 48-well plate at a density of 6 × 10^4^ cells per well and incubated for 24 h in the complete culture medium. Then, AAP PLGA NPs (10 mM) were added, and the cells were cultured for another 24 h. The original culture medium was removed, and the cells were washed twice with PBS. Live/dead cell double staining reagents (calcein AM and propidium iodide) were added, and the cells were incubated at 37 °C in the dark for 20 min. After incubation, the staining solution was removed, and the cells were washed three times with PBS and imaged using a fluorescence microscope, with live cells appearing green and dead cells appearing red. Randomly selected areas were photographed for live/dead cell quantitation. Three replicate controls were used for each group.

### 2.8. In Vivo Fluorescence Imaging

Eighteen healthy C57/BL6 male mice aged 8 weeks with a body weight of 21 ± 2 g were allowed to fast for 12 h and administered ICG-labeled AAP PLGA NPs (20 μL) through the nasal cavity. Drug circulation was monitored using a small animal in vivo fluorescence imaging system (excitation at 785 nm, emission at 810 nm) at different time intervals (0, 0.5, 1.5, 3, 6, and 24 h).

### 2.9. Determination of AAP PLGA NPs in the Murine Blood and Brain

Forty-eight healthy C57/BL6 male mice aged 8 weeks with a body weight of 21 ± 2 g were randomly divided into two groups after 12 h fasting. AAP PLGA NPs and AAP sol (20 µL) were administered intranasally, and blood (~0.5 mL) was collected from eye sockets at 0.5, 1, 1.5, 2, 4, 6, 12, and 24 h. The blood samples were placed in heparin-washed centrifuge tubes and centrifuged at 4000 rpm for 3 min, and the upper plasma layer was collected. After blood collection, the animals were euthanized and the cervical vertebrae were rapidly dissected. The brain tissue was removed, washed with physiological saline, dried with filter paper, weighed, and homogenized with physiological saline. The absorbance of the resulting supernatant (or blood plasma) at 257 nm was determined using UV–visible spectroscopy and converted into acetaminophen concentration according to the working curve method.

### 2.10. Pharmacodynamic Study of AAP PLGA NPs

#### 2.10.1. Hot-Plate Test [14]

The experiment was performed at an ambient temperature of 22 °C. The hot plate temperature was adjusted to 55 ± 0.5 °C. The left and right hind paws of 8-week-old healthy C57/BL6 male mice were placed flat on the center of the hot plate to ensure sufficient contact with the soles. The time elapsed until the first occurrence of foot licking was taken as the latency period (pain threshold), and a second measurement was performed after 5 min. The average of these two measurements was used as the baseline pain threshold. To avoid scalding, the test was stopped after 60 s. Qualified mice with a basic pain threshold of 5–30 s and no jumping responses were selected.

#### 2.10.2. Tail-Flick Test [15]

During the experiment, the room temperature was controlled at ~22 °C, and the radiation power was set to 34 W. Healthy 8-week-old C57/BL6 male mice were placed in a mouse fixator, and their tails were exposed to light. The irradiation site was fixed at the middle and lower one-third of the tail, and the experiment was started after the mice had calmed. The time elapsed from the start of irradiation to the occurrence of the tail flick reaction was recorded as the pain threshold. The measurement was repeated after 5 min, and the average of the two reactions was taken as the baseline pain threshold. To protect the tail from injury, mice that did not flick their tail for more than 10 s were assigned a pain threshold of 10 s. Qualified mice with a basic pain threshold of 3–6 s were selected.

#### 2.10.3. Model Establishment

Forty qualified mice were selected based on the results of the hot plate and tail flick experiments and randomly divided into four groups with 10 mice each. In the control (C) group, the mice were placed under normal pressure and oxygen conditions without any treatment. In the rapid high-altitude (H) group, the mice were placed in a low-pressure oxygen chamber simulating a high-altitude environment (6000 m) for 7 days to establish a model. In the AAP sol treatment (AH) group, the mice were placed in the low-pressure oxygen chamber (6000 m) for 7 days and given 20 µL of the AAP sol every morning between 8:00 and 9:00. In the AAP PLGA NP treatment (ANH) group, the mice were placed in the low-pressure oxygen chamber (6000 m) for 7 days and given 20 µL of AAP PLGA NPs between 8:00 and 9:00 every day (Figure 2A).

All mice were subjected to hot plate foot licking and tail flick experiments at the same time (14:00–18:00) on days 1 (T1), 2 (T2), 3 (T3), 4 (T4), 5 (T5), 6 (T6), and 7 (T7) after modeling, and the pain threshold was recorded.

### 2.11. Statistical Analysis

Data were processed using GraphPad Prism 8.0 software. The results were expressed as means ± standard deviations using one-way analysis of variance or the independent samples *t*-test. The statistical significance level was set to *p* < 0.05.

## 3. Results

### 3.1. Properties of AAP PLGA NPs

The PLGA and AAP PLGA NPs were spherical and had uniform size distributions (Figure 1B), with the respective average sizes/PDIs/zeta potentials determined as 106 nm/0.243/−24.1 mV and 118 nm/0.221/−23.7 mV (Figure 1C). Nanoparticle-containing systems are typically stable and not prone to precipitation or flocculation at |zeta potential| ≥ 20 mV and PDI < 0.3. These conditions were met in the case of AAP PLGA NPs, and the corresponding formulation was therefore concluded to be stable. The EE and DL were determined as 42.53% and 3.87%, respectively.

### 3.2. In Vitro Release Behavior and Stability

The amount of acetaminophen released in SNF was lower than the amounts released in PBS and CSF in the first 2 h (Figure 1D). The cumulative release of the drug in PBS at 2, 4, 6, and 8 h was 35%, 56%, 64%, and 66%, respectively. The cumulative release of drugs in SNF at 2, 4, 6, and 8 h was 13%, 51%, 73%, and 77%, respectively. Cumulative release rates of ~69%, 82%, and 81% were observed in PBS, SNF, and CSF, respectively, after 12 h, and the percentage of drug release remained constant until 72 h. This release phenomenon is consistent with the biphasic pattern of drug release in PLGA matrix, which is characterized by sudden release followed by sustained release [16]. The better release characteristics observed in SNF and CSF (compared with those observed in PBS) indicated that the formulation could be effectively delivered to the brain via the intranasal route. Additionally, continuous drug release over 72 h allowed the effectiveness of the drug to be maintained for a longer period. Over time, AAP PLGA NPs maintained relatively stable particle size, PDI, and Zeta potential in three different media (Figure 1E). The pH value of SNF is 5.5–6.5. In this medium, although the surface potential of nanoparticles exhibited negative charge, its absolute value was slightly lower compared to PBS and CSF. The possible reason is that H^+^ in SNF adsorbs on the surface of nanoparticles, causing a decrease in the absolute value of the detected surface potential of nanoparticles [17]. At the same time, the changes in particle size and PDI of nanoparticles in SNF were more significant, indicating that PLGA nanoparticles degrade faster in a slightly acidic environment.

### 3.3. Results of Cytotoxicity Assessment

The absence of significant viability changes in the placebo, AAP PLGA NP, and AAP sol groups with the increasing treatment concentration indicated that this concentration had no marked effect on the viability of the HT22 cells (Figure 3A). The concentration of AAP PLGA NPs was selected as 10 mM. Compared with that in the placebo group at the same concentration, the cell viabilities in the AAP PLGA NP and AAP sol groups (>97% in both cases) showed no significant differences. This indicated that the impact of the nanomaterials and drugs on cells was very small, i.e., the drug delivery system was highly biocompatible.

### 3.4. Live/Dead Staining

The number of dead cells in the AAP PLGA NP group was not significantly different from that in the control group, and the field of view was mainly occupied by live cells (Figure 3B,C).

### 3.5. Live Fluorescence Imaging

The results of live fluorescence imaging indicated that the intranasally administered ICG-labeled AAP PLGA NPs mainly targeted the brain (Figure 4).

### 3.6. Drug Levels in Murine Blood and Brain

Compared with the AAP sol group, the AAP PLGA NP group showed a linear decrease followed by an increase in blood and brain drug concentrations over time after intranasal administration. The blood and brain drug concentrations in both groups first increased and then decreased with time. These results indicated that the AAP PLGA NPs exerted a slow-release effect after intranasal administration. Further details are shown in Figure 5A,B.

### 3.7. Pharmacodynamic Observation

Compared with the AAP sol group, the AAP PLGA NP group showed a significantly increased hot-plate pain threshold and tail flick latency in mice rapidly entering high altitudes (Figure 2B,C).

## 4. Discussion

This study focuses on addressing the first-pass liver elimination problem and potential risk of liver and kidney damage associated with oral or intravenous administration of acetaminophen (AAP). An innovative dosing strategy is proposed, which involves loading acetaminophen into PLGA nanoparticles and administering it intranasally to mice rapidly entering high-altitude environments. This strategy aims to increase the brain concentration of drugs, prolong their efficacy, and reduce systemic side effects, providing a new solution for pain management in high-altitude environments.

PLGA-based nanodrug delivery systems are typically prepared using emulsification/solvent evaporation [18], nanoprecipitation [19], self-assembly [20], double emulsification [21], or supercritical fluid technologies [22]. The combination of ultrasonication-assisted emulsification and solvent evaporation in the preparation of PLGA-based nanomedicine delivery systems allows one to increase the EE and DL, control the nanoparticle size and morphology, simplify the preparation process, increase the controllability of drug release, and ensure applicability to multiple delivery pathways [18]. Therefore, in the process of preparing AAP PLGA NPs, we employed a combination of ultrasonic emulsification and solvent evaporation, which not only ensured the uniformity and stability of the nanoparticles, but also improved the encapsulation efficiency and drug loading capacity of the drugs. Specifically, the drug encapsulation efficiency of 42.53% and drug loading capacity of 3.87% indicate that our preparation process could efficiently encapsulate drugs, laying a solid foundation for subsequent pharmacological research [23,24]. The morphology (spherical) and polydispersity index (0.221) of the AAP PLGA NPs favored their uniform distribution on the nasal mucosa, which was expected to result in an increased area of contact with the same and thus improve the drug absorption efficiency [25]. Previous studies have shown that 100–200 nm [26] NPs can pass through the mucus layer in the nasal cavity, reach the brain through the olfactory nerve axon, and avoid rapid clearance by nasal mucosal cilia, thereby shortening the onset of action of the encapsulated drug. The in vivo fluorescence imaging results showed that AAP PLGA NPs labeled with ICG with a size of 118 nm could reach the animal brain in a short period of time after intranasal administration, which is consistent with previous studies. From the stability test results, it can be seen that the strong negative charge (−23.7 mV) on the surface of the NP could maintain good stability in different pH media by preventing aggregation and precipitation [27]. In addition, the results of the cytotoxicity test also indicate that the strong negative charge on the surface of NP helped hinder non-specific interactions with nasal mucosal cells, thereby reducing potential immunogenicity and cytotoxicity [28]. This provides a strong guarantee for its safety in clinical applications in the future. In vitro release experiments showed that the AAP PLGA NPs could continuously release acetaminophen for up to 72 h (in PBS, SNF, and CSF), with the release rate reaching ~82%. The sustained release properties of AAP PLGA NPs resulted in a prolonged drug release time in vivo. Compared with traditional acetaminophen formulations, intranasally administered AAP PLGA NPs maintained effective drug concentrations for a longer period of time, thus having the potential to reduce the dependence on frequent dosing and improve medication adherence [29]. Analysis of drug concentrations in blood and brain showed that compared to AAP sol, AAP PLGA NPs showed a significant increase in drug levels in the brain of mice after nasal administration, with delayed elimination. This discovery not only validates the effectiveness of our drug delivery strategy, but also reveals its enormous potential in increasing drug concentration in the brain and prolonging drug efficacy.

Furthermore, we evaluated the effect of AAP PLGA NPs on the pain threshold of mice rapidly entering high-altitude environments through the hot-plate test and tail-flick test [30]. The results showed that compared with AAP sol, AAP PLGA NPs administered via the nasal route significantly increased the pain threshold of mice. This discovery indicates that our dosing strategy not only increases the concentration of drugs in the brain, but also achieves effective management of pain, especially suitable for pain relief in extreme environments such as high altitude.

## 5. Conclusions

In this study, we used a combination of ultrasonic emulsification and solvent evaporation to prepare nanoparticles loaded with acetaminophen. These nanoparticles exhibited good stability, safety, and sustained release properties. More importantly, when administered through the nasal cavity, these nanoparticles could quickly reach the brain, effectively increasing the pain threshold of mice in high-altitude environments. Looking ahead, we will further explore the potential applications of AAP PLGA NPs in other pain management scenarios, such as chronic pain and postoperative pain. Meanwhile, we will also optimize the preparation and characterization methods of nanoparticles to improve their therapeutic efficacy and safety. In addition, we will conduct in-depth research on the impact of special physiological changes in high-altitude environments, such as hypoxia and low air pressure, on the efficacy of AAP PLGA NPs, in order to provide a scientific basis for their precise medication in clinical applications.

## Figures and Tables

**Figure 1 pharmaceutics-17-00341-f001:**
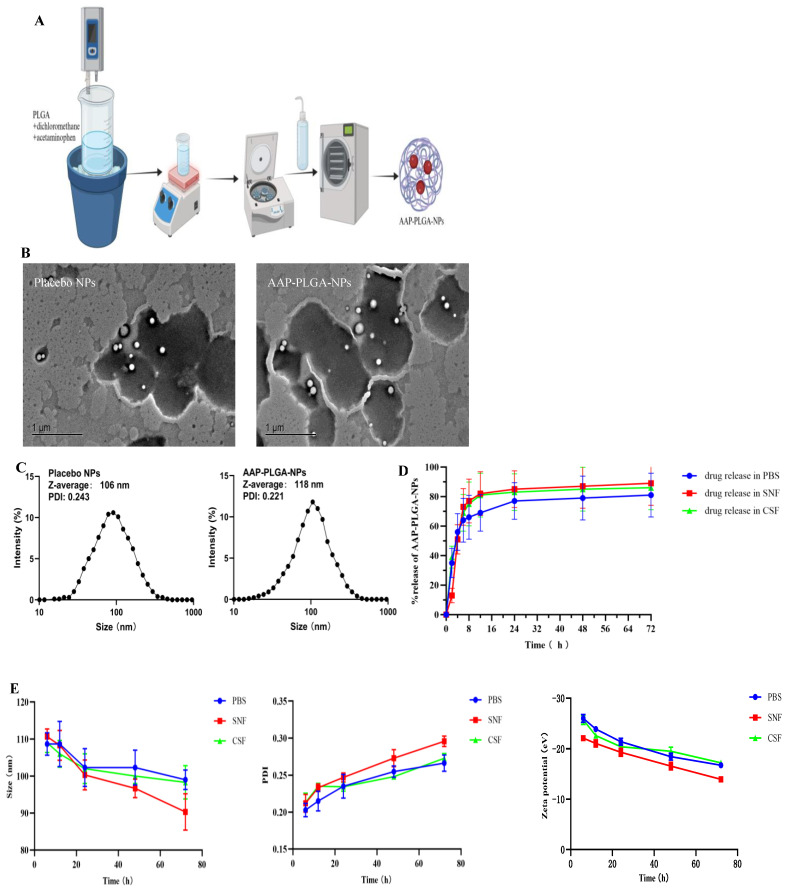
Preparation and characterization of placebo NPs and AAP PLGA NPs. (**A**) Preparation process of nanoparticles. (**B**) TEM images of placebo NPs and AAP PLGA NPs, Scale bar: 1 μm. (**C**) The particle size distribution of placebo NPs and AAP PLGA NPs. (**D**) The release of AAP from AAP PLGA NPs in different media. Data are presented as mean ± SD (n = 3). (**E**) The changes in particle size, PDI, and zeta potential of AAP PLGA NPs over time in different media. Data are expressed as mean ± standard deviation.

**Figure 2 pharmaceutics-17-00341-f002:**
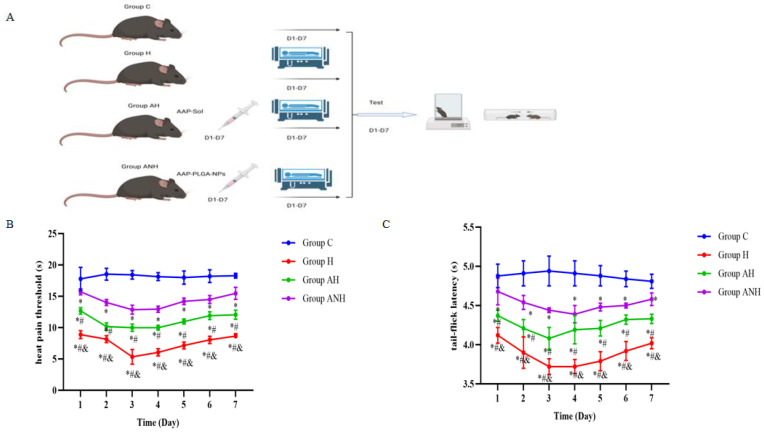
The effect of AAP PLGA NPs on the pain threshold of mice rapidly entering high altitude. (**A**) Experimental timeline: Mice were given either AAP PLGA NPs or AAP sol via nasal route prior to modeling, and the potential effects of the drug on their pain threshold were evaluated using the hot plate experiment and tail flick experiment from the first to the seventh day after entering the plateau. (**B**) Comparison of heat pain thresholds. Data are presented as mean ± SEM of 10 animals per group. * *p* < 0.05, # *p* < 0.05, & *p* < 0.05; One-way ANOVA. (**C**) Comparison of tail-flick latency. Data are presented as mean ± SEM of 10 animals per group. * *p* < 0.05, # *p* < 0.05, & *p* < 0.05; One-way ANOVA.

**Figure 3 pharmaceutics-17-00341-f003:**
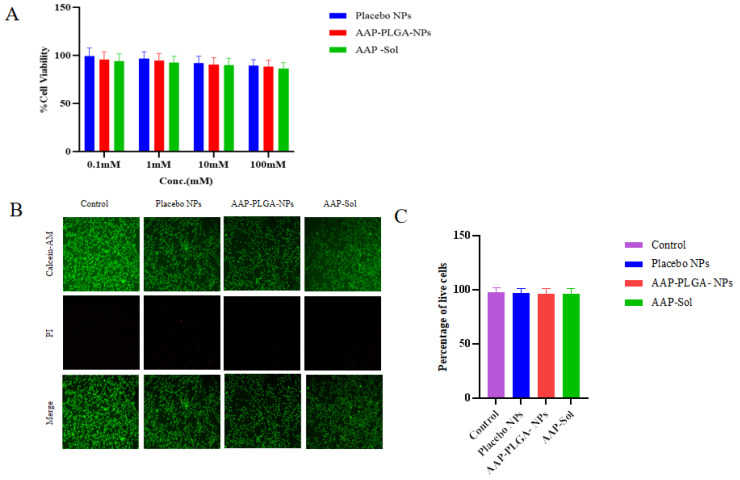
Biocompatibility determination of nanomedicine. (**A**) Cytotoxicity of placebo NPs, AAP PLGA NPs and AAP sol against HTT-22 cells. Data are presented as mean ± SD (n = 3). (**B**) Live/dead cell staining images of HTT-22 cells treated with placebo NPs, AAP PLGA NPs and AAP sol. Scale bar: 500 μm. (**C**) Statistical analyses of live/dead cell staining of HTT-22 cells. Data are presented as mean ± SD (n = 3). One-way ANOVA.

**Figure 4 pharmaceutics-17-00341-f004:**
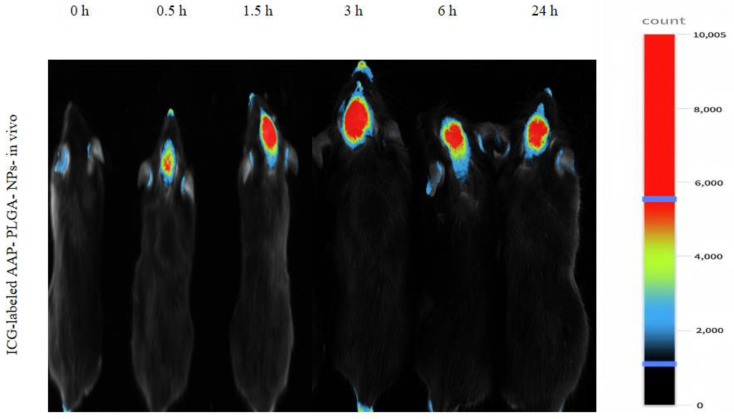
Fluorescence imaging and in vivo distribution of ICG-labeled AAP PLGA NPs.

**Figure 5 pharmaceutics-17-00341-f005:**
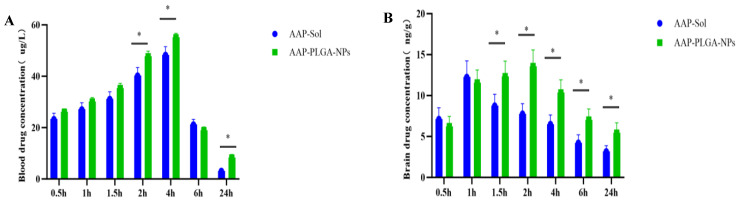
The drug concentration in blood and brain tissue after nasal administration. (**A**) Drug concentrations in the blood at different time points after intranasal administration of AAP sol and AAP PLGA NPs. Data are presented as mean ± SD (n = 3). * *p* < 0.05; independent-samples *t*-test. (**B**) Drug concentrations in the brain at different time points after intranasal administration of AAP sol and AAP PLGA NPs. Data are presented as mean ± SD (n = 3). * *p* < 0.05; independent-samples *t*-test.

## Data Availability

The data presented in this study are available from the corresponding author upon reasonable request.

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
