# Peer review of "Intranasal Administration of Acetaminophen-Loaded Poly(lactic-co-glycolic acid) Nanoparticles Increases Pain Threshold in Mice Rapidly Entering High Altitudes"

_pharmaceutics, 2025, doi:10.3390/pharmaceutics17030341_

Round 1
Reviewer 1 Report
Comments and Suggestions for Authors
Pharmaceutics-3468086
Intranasal Administration of Acetaminophen-Loaded Poly(Lactic-co-Glycolic Acid) Nanoparticles Increases Pain Threshold in Mice Rapidly Entering High Altitudes
Qingqing Huang, Xingyue Han, Jin Li, Xilin Li, Xin Chen , Jianwen Hou, Sixun Yu, Shaobing Zhou, Gu Gong Haifeng Shu
In the presented study authors aimed to develop an efficient method of pain management in high-altitude environments. They explored the effects of intranasally administered acetaminophen-loaded PLGA nanoparticles (AAP PLGA NPs) on the pain threshold in mice rapidly entering high altitudes. In the work authors describe the production of nanoparticles, characterize them using a number of physicochemical methods and make the transition from in vitro testing to in vivo experiments.
This theme is in the scope of Pharmaceutics. The data obtained are of interest for researchers.
However, I have a number of questions and comments.
I think the discussion section should be rewritten, making it more extensive, comparing the data obtained with previous studies. The beginning of this section is very general and would be more appropriate in the introduction.
Did the authors provide an optimal method for obtaining drug-loaded nanoparticles? Were other protocols tested that varied the particle formation conditions? What is the structure of the resulting nanoparticles?
The authors should confirm the stability of the obtained particles. Perhaps the DLS method can be useful for this, with the help of which it is possible to track the change in the size and charge of the particles over time.
Probably, the negative zeta potential of nanoparticles is associated with partial dissociation of polyacids on their surface. Will the zeta potential of particles change with variations in the pH of the environment. Will the particles remain stable in this case?
Figure 1 should provide data characterizing the release of the unformulated drug.
The phrase “Compared with traditional acetaminophen formulations, intranasally administered AAP PLGA NPs maintained effective drug concentrations for a longer period of time, thus having the potential to reduce the dependence on frequent dosing and improve medication adherence” (lines 351-353) requires confirmation by literature data.
The conclusion section is not mandatory for the Pharmaceutics. But it would be useful, as it allows authors to summarize the research conducted and indicate prospects for further development of the work.
Minor comments:
Section 2.2 should indicate the purity of the drugs and reagents used
The pore diameter of the dialysis bags used should be indicated
More detailed information on the measurements carried out by the DLS method (a scattering angle, a number of measurements for each sample, etc.) should be given. What does the term "size" mean? We are talking about the hydrodynamic radius or diameter?
Figure 1 should be removed in Part 3 (Results).
Authors write, “In vitro release experiments showed that the AAP PLGA NPs could continuously release acetaminophen for up to 72 h …, with the release rate reaching 82%” (lines 349). In fact, it is not about the speed, but about the degree (or percentage) of drug release.
Taking into account the above, I believe that manuscript needs major revision before it can be recommended for publication.
Author Response
请参阅附件。

Reviewer 2 Report
Comments and Suggestions for Authors
The authors have conducted significant research on the intranasal delivery route for the sustained release of a commonly used analgesic in high-altitude conditions, as presented in the article "Intranasal Administration of Acetaminophen-Loaded Poly(Lactic-co-Glycolic Acid) Nanoparticles Increases Pain Threshold in Mice Rapidly Entering High Altitudes." The study is well-designed and executed; however, additional examples and references in the discussion are needed to support the observed results.
Comments/Suggestions:
- Line 142-144: What is the rationale behind using 20 mL of release media? Is it representative of CSF/SNF? Please discuss or mention that the release study was performed as a proof-of-concept for release.
- Line 251-254: The drug release from the PLGA matrix is typically biphasic, with an initial burst followed by sustained release, as reported in the literature. Can you provide a reference to support the claim of initial slow release?
- Line 339-342: Did you perform nanoparticle stability studies? Please include the results. The statement in the discussion needs references to relevant articles to support the claim of nanoparticle stability.
- Line 304-307: What other pharmacodynamic observations are used to evaluate pain levels in mice? What is the significance of the Hot-Plate and Tail-Flick tests compared to others?
- Discussion: References to relevant examples from the literature regarding these studies/results are missing and need to be added to support the claims made in the discussion section.
Author Response
请参阅附件.

Round 2
Reviewer 1 Report
Comments and Suggestions for Authors
The authors took into account the reviewer’s comments and made corrections to the new version of the article. The revised version may be recommended for publication.
Author writes in his responses to the reviewer: “May I ask if all the images in Figure 1 should be removed from the results, or only one of them should be deleted. In addition, where should the deleted images be placed in the article. I don't quite understand the reviewer's opinion on this issue.” In my opinion Fig. 1 presents the data obtained during the research. Therefore, I propose to move the figure and explanations to it from the experimental part to the part results. Perhaps the description of the presented data could be given as a separate part at the beginning of Section 3. However, if the editors consider acceptable to leave the original placement of the figure, I will not insist on my proposal.